# Double Hydrogen Bonding between Side Chain Carboxyl Groups in Aqueous Solutions of Poly (β-L-Malic Acid): Implication for the Evolutionary Origin of Nucleic Acids

**DOI:** 10.3390/life7030035

**Published:** 2017-08-28

**Authors:** Brian R. Francis, Kevin Watkins, Jan Kubelka

**Affiliations:** 1Department of Molecular Biology, University of Wyoming, Laramie, WY 82071, USA; 2Department of Chemistry, University of Wyoming, Laramie, WY 82071, USA; kwatkin6@uwyo.edu (K.W.); jkubelka@uwyo.edu (J.K.)

**Keywords:** poly (β-L-malic acid), double hydrogen bonding, intramolecular, nucleic acid evolution

## Abstract

The RNA world hypothesis holds that in the evolutionary events that led to the emergence of life RNA preceded proteins and DNA and is supported by the ability of RNA to act as both a genetic polymer and a catalyst. On the other hand, biosynthesis of nucleic acids requires a large number of enzymes and chemical synthesis of RNA under presumed prebiotic conditions is complicated and requires many sequential steps. These observations suggest that biosynthesis of RNA is the end product of a long evolutionary process. If so, what was the original polymer from which RNA and DNA evolved? In most syntheses of simpler RNA or DNA analogs, the D-ribose phosphate polymer backbone is altered and the purine and pyrimidine bases are retained for hydrogen bonding between complementary base pairs. However, the bases are themselves products of complex biosynthetic pathways and hence they too may have evolved from simpler polymer side chains that had the ability to form hydrogen bonds. We hypothesize that the earliest evolutionary predecessor of nucleic acids was the simple linear polyester, poly (β-D-malic acid), for which the carboxyl side chains could form double hydrogen bonds. In this study, we show that in accord with this hypothesis a closely related polyester, poly (β-L-malic acid), uses carboxyl side chains to form robust intramolecular double hydrogen bonds in moderately acidic solution.

## 1. Introduction

How life started on Earth is one of science’s most confounding problems [1]. Two major difficulties are encountered when proposing a solution. First, other than very low levels of oxygen in the atmosphere, little information is available concerning the conditions on the Earth during the Hadean eon when life is thought to have emerged [2], which means that there is a wide spectrum of possible conditions. Secondly, biochemistry as found in modern cells is far too complicated to have spontaneously emerged under prebiotic conditions [3]. Proteins, which are involved in almost every aspect of cellular activity, have a simple peptide backbone but their biosynthesis is an extremely complicated process catalyzed by other proteins, RNAs, and RNA/protein complexes. Together they permit incorporation of a repertoire of up to 22 amino acids, some of which have chemically complex side chains [4]. A reasonable assumption is that proteins may at one time have been synthesized using a simpler process from a small number of amino acids with simple side chains. By contrast, RNA and DNA are complicated in both the backbones and the side chains. Biosynthesis of RNA and DNA involves long synthetic pathways catalyzed by numerous enzymes [5]. A simple evolutionary predecessor of the genetic polymers is not immediately obvious. The difficulty in understanding the events that led to the presence of RNA and DNA in biology is a major reason why an explanation for the origin of life has proven so elusive.

Two different approaches have been taken to explain the origin of life. One approach, the widely accepted RNA world hypothesis, contends that RNA was both the original genetic polymer and the original catalytic polymer, preceding DNA and proteins [6,7,8]. This hypothesis is part of the ‘genetics first’ approach to the origin of life. Catalysts made from RNA were involved in both replication and metabolism. However, how such a complicated polymer as RNA was first produced on the primitive Earth is a major problem for this hypothesis, even for those who generally accept it [9]. Chemical synthesis of RNA under presumed prebiotic conditions requires many sequential steps [10]. Polymers have been synthesized that have simpler backbones than RNA while retaining purines and pyrimidines for base pairing [11,12], but these polymers are still complicated. In the other approach to the origin of life, described as ‘metabolism first’, a simple metabolism such as the reverse citric acid cycle or the acetogenic pathway was established prior to the appearance of genetic or catalytic polymers [13,14,15,16,17,18,19,20]. While this approach involves molecules that are more feasible for the early Earth, it does not adequately address the subsequent appearance of genetic polymers. Consequently, neither of these two approaches presents a plausible explanation for the production of such a complex molecule as RNA from a mixture of small molecules on the primitive Earth.

An alternative proposal is that in concurrence with its complexity RNA had an extensive evolutionary history, and that the evolutionary process began inside primitive cells that coupled redox energy to activation of carboxylic acids as thioesters and the synthesis of polyesters and polypeptides [3,21,22]. If this were the case, what was the earliest evolutionary predecessor of RNA? Because a genetic system would be needed early in the emergence of life in order for selection to occur for improved fitness, the early genetic polymer must have been simple. It would have needed a simple backbone and side chains that had the ability to form two roughly parallel hydrogen bonds. The simplest functional groups with this ability are carboxyl and carboxamide. In particular, exceptionally strong cyclic double hydrogen bonds are known to form between carboxyl groups [23]. To form hydrogen-bonded structures similar to those in nucleic acids, the carboxylic side-chains had to occur at regularly repeating distances along the polymer backbone and point in the same direction relative to the backbone.

Accordingly, our hypothesis is that nucleic acids evolved from a molecule with the above characteristics: poly (β-d-malic acid). In addition to its relative simplicity, the building block—malic acid—is a component of carboxylic acid cycles found at the center of biochemical metabolism [5] and it is therefore highly likely to have been produced early in metabolic evolution [3]. Thus, this hypothesis also effectively reconciles the ‘genetics first’ and ‘metabolism first’ approaches.

Although poly (β-D-malic acid) itself is not an information polymer because all of the side chains are the same, it could have become an information polymer by modification of the side chains with another carboxyl-containing molecule [3]. Sequences of modified and unmodified side-chains could constitute the first primitive genetic alphabet (Figure 1). D-malic acid is involved rather than L-malic acid because there is a plausible evolutionary pathway leading to D-ribose based RNA and DNA [3]. Specifically, the three carbons in the backbone of poly (β-D-malic acid) evolved into the 3′, 4′, and 5′ carbons of D-ribose. Replacement of the C4 carboxyl group of malic acid by a hydroxymethyl group allowed linking by phosphate. The carboxyl side chains evolved into the pyrimidines and purines. The first step involved replacement of the carboxyl groups by carboxamide groups that are not ionized at about neutral pH. Because biosynthesis of aromatic molecules starts with aliphatic molecules, the evolutionary pathway from the amide side chains to the purines and pyrimidines involved side chain extension, cyclization, and oxidation (for details, see [3]). Obviously, this hypothesis requires considerable experimental testing and this report is the first step in this direction.

Poly (β-D-malic acid) is not produced biologically but the closely related polymer poly (β-L-malic acid) (PBMA) (Figure 2a) would have almost identical properties to poly (β-D-malic acid) and is produced by microorganisms such as *Physarum polycephalum* and *Aureobasidium pullulans* [24], where it performs functions that are not genetic [25]. Previous studies of solid PBMA show the presence of hydrogen bonding between carboxylic acid side chains [26]. It has only recently been shown that simple carboxylic acids, such as acetic and lactic acids, exist predominantly as cyclic hydrogen bonded dimers in aqueous solution rather than in other hydrogen bonded forms that can also involve water [27,28]. Therefore, study of PBMA in aqueous solution is expected to yield important evidence supporting the poly (β-D-malic acid) hypothesis. In particular, the presence of double hydrogen bonds between side chain carboxyl groups could be established that are critical for formation of nucleic acid-like structures. In this report, we show for the first time that a simple polymer, PBMA, forms intramolecular double hydrogen bonds between carboxyl side chains in moderately acidic solution, similar to the base pairing found in RNA. In addition, we provide initial structural data on PBMA, and demonstrate that ester bonds are stable under the conditions that allow the double hydrogen bonds to form.

## 2. Experimental Section

Samples for Fourier Transform infrared spectroscopy (FT-IR) were dissolved in either water or a buffer, loaded in a CaF_2_ disc with a 5 μm indentation that created a 6 μL volume (Biocell, BioTools, Jupiter, Florida, USA), and set in the beam of a Bruker Tensor 27 FT-IR instrument operated by OPUS 5.5 software (Opus 5.5). Spectra were obtained from an average of 256 scans of the sample taken from 4000 to 700 cm^−1^. To produce the spectrum of malic acid in D_2_O, malic acid was twice dissolved in D_2_O and dried under vacuum, and then dissolved in D_2_O prior to measurement. Most absorption values were obtained after subtraction of the absorption of water.

Liquid chromatography employed a size exclusion HPLC column (Bio-Rad Bio-sil SEC 250-5) attached to an Amersham FPLC machine. PBMA dissolved in 100 mM potassium phosphate at pH 2.3 or 7.2 and 10 mg/mL (90 mM in side chains) was loaded in a 100 μL loop and eluted with the corresponding buffer at 1.0 mL/min with detection of eluted material using UV light at 210 nm.

Circular dichroism (CD) spectroscopy was performed on an Aviv Model 430 spectropolarimeter using a solution of 0.1 or 1.0 mg/mL PBMA (0.9 or 9 mM in side chains) in 100 mM potassium phosphate buffer at pH 2.2 or 7.2. The sample was placed in a quartz cuvette with a 1 mm path length and four scans between 260 and 180 nm were collected and averaged. The intensity is plotted as the molar extinction coefficient per mole of carboxylic side-chains.

Quantum chemical calculations were performed at density functional theory (DFT) level using Gaussian 09 Rev. D.01 software package [29]. All structures were optimized with B3LYP density functional [30] and 6-31+G(d,p) basis set [31,32]. The conductor-like polarized solvent model (CPCM) [33] was used in all calculations to approximate the aqueous solvent. Anharmonic vibrational spectra calculations were performed using 2nd order vibrational perturbation theory [34,35] at the same level. CD spectra simulations were carried out using time-dependent DFT (TD-DFT) with the range separated CAM-B3LYP density functional [36] and the same 6-31+G(d,p) basis set [31,32]. The CD spectral contours were constructed by assigning a Gaussian shape with full width at half maximum (FWHM) of 25 nm and summing.

Atomic force microscopy (AFM) was performed using a solution of 0.5 mg/mL PBMA in water and freshly cleaved mica sections. Either mica was dipped in PBMA solution and quickly dried or 5 μL of solution was dropped onto mica and dried. AFM was performed in air with an Asylum Instruments Cypher in AC imaging mode.

## 3. Results

### 3.1. Lactic and Malic Acids Have Double Hydrogen Bonds between Carboxyl Groups in Moderately Acidic Solutions

Exceptionally strong cyclic hydrogen bonding between carboxyl groups is explained by resonance with an ionic structure [23] (Figure 2b) and is often detected by infrared (IR) spectroscopy as an absorption peak for O-H stretching in the region 3300–2500 cm^−1^, compared to ~3500 cm^−1^ for the monomer, and a peak for C=O stretching at 1720−1706 cm^−1^, compared to ~1760 cm^−1^ for the monomer. Non-aqueous solvents are generally used for solution IR spectroscopy of organic molecules because strong water absorption in the ~3400, ~2100, ~1650, and ~800 cm^−1^ regions obscure signals of functional groups including those arising from O-H and C=O stretching. If the path length is low (less than ~6 μm) and the concentration of organic molecules is high (≳0.5 M), absorbance by solutes can be observed in aqueous solution. Additional techniques, including vibrational circular dichroism and ultrafast time-domain Raman spectroscopy, have been used to show that lactic and acetic acids exist primarily as cyclic dimers in water [27,28]. We used spectroscopic and other techniques to characterize the hydrogen bonding between carboxyl groups of PBMA and other carboxylic acids in aqueous solution. The results are shown in Figure 3 and the complete list of relevant IR peak frequencies is given in Table 1.

Prior to FT-IR measurements of PBMA in aqueous solution, comparison data were obtained for monomeric malic acid, as well as lactic and acetic acids that are known to form dimers in solution [27,28], and therefore served as useful controls. The spectra of lactic acid (Figure 3a) and malic acid (Figure 3b) were measured, with that of malic acid being similar to that previously reported [37]. At 2 M concentration in water and 23 °C, lactic and malic acid (pH 2) had C=O stretching peaks, respectively, at 1728 and 1724 cm^−1^, characteristic of carboxylic acid dimers, which partially overlapped with a 1644 cm^−1^ band due to water bending. That the latter corresponds to the incompletely subtracted H_2_O bending signal was confirmed by the spectra of malic acid dissolved in D_2_O rather than H_2_O. In D_2_O, the bending mode appears near 1210 cm^−1^, and the C=O stretching region contains only a single band at 1718 cm^−1^ (Table 1).

Some of the O-H stretching absorbance of malic acid was subsumed under the strong absorption of water centered around ~3400 cm^−1^, but a band with a maximum at 2600 cm^−1^ within the O-H stretching region for carboxylic acids indicated strong double hydrogen bonding (Figure 3, Table 1). When malic acid was dissolved in 100 mM potassium phosphate at pH 2.2 the same spectrum was obtained, but at pH 7 the 1724 cm^−1^ C=O stretching band was no longer present, while additional peaks appeared at 1572 and 1397 cm^−1^ characteristic of carboxylate (COO^−^) stretching. The absorption at ~2600 cm^−1^ observed in acidic solution was not present, confirming that it was due to carboxylic acid O-H stretching (Figure 3).

These assignments were further confirmed by quantum chemical, density functional theory (DFT) computations of vibrational spectra. The results are summarized in Table 2. The emergence of the O-H stretching modes of doubly hydrogen-bonded COOH red-shifted by ~800–1000 cm^−1^ compared to non-hydrogen-bonded COOH groups was clearly apparent from the simulations. Although the (unscaled) harmonic frequencies were systematically too high, anharmonic vibrational calculations predicted these modes to be near 2600 cm^−1^, in excellent agreement with 2600 cm^−1^ observed experimentally. The calculations also showed a red shift in C=O stretching frequencies upon hydrogen bonding, as expected, but both harmonic and anharmonic approximations underestimated the experimental values. Note that the in-phase C=O stretch of the doubly H-bonded COOH groups, denoted by a (+) in Table 2, had near zero intensity and was not experimentally observed. Substances that can form multiple single hydrogen bonds (20% glycerol, glucose, and galactose) did not significantly alter the spectrum of malic acid (Table 1). These results demonstrated that malic acid in acidic solution had double hydrogen bonds between carboxyl groups, which may be similar to those observed in crystalline malic acid where both carboxyl groups form double hydrogen bonds [38]. Double hydrogen bonding was also observed for lactic and acetic acids, confirming previous reports [27,28], and β-alanine (Table 1), which will be discussed later with regard to possible synthesis of a simple information polymer.

### 3.2. PBMA Has Stronger Double Hydrogen Bonds than Malic and Lactic Acids

PBMA (generously provided by José Portilla-Arias) had an average molecular mass of ~40 kDa, corresponding to an average length of ~350 malic acid residues per strand, and a purity of 99.65%, with the minor impurity being RNA/DNA. PBMA dissolved in water at 232 or 464 μg/μL had a pH of 1.5–2 and concentrations of 2 M and 4 M in carboxylic acid side chains. High concentrations of PBMA were used to demonstrate the presence of cyclic double hydrogen bonds, not because they represent concentrations expected in primitive cells. Nevertheless, the concentration of nucleotides in RNA in an *Escherichia coli* cell (0.1 pg RNA per 1 μm^3^ cell) is ~0.3 M. FT-IR spectra at 23 °C of PBMA (Figure 3c) showed a broad absorbance band centered at 2550 cm^−1^ characteristic of dimeric carboxylic acid O-H stretching that was similar to the corresponding peaks for 2 M and 4 M malic acid, but its lower frequency indicated stronger hydrogen bonding.

The C=O stretching peak of PBMA observed at 1742 cm^−1^ contained overlapping bands from both side-chain carboxylic groups and backbone ester groups. The absorption of the latter appeared slightly lower, at 1736 cm^−1^, as evident from the neutral pH spectra (Figure 3c) where the doubly hydrogen-bonded COOH signals vanish. The side-chain C=O band was higher than the ~1720 cm^−1^ frequency generally observed for a carboxyl group [23]. A single C=O stretch at 1735 cm^−1^ was also reported for solid PBMA [26]. Higher C=O stretching frequency compared to lactic and malic acids (Figure 3, Table 1) would be consistent with reduced hydrogen bonding to surrounding water [39] and stronger double hydrogen bonds between the PBMA carboxylic groups, as evidenced by the red shift of O-H stretching discussed in the previous paragraph. However, other effects may also play a role causing the blue-shift of this band. In particular, as the circular dichroism spectra and theoretical modeling (see Figure 7, below) suggest, the PBMA strands may have sections of turns or loops in which the carboxylic acid side chains are not hydrogen bonded to other carboxyl groups. These unpaired groups may contribute higher vibrational frequencies, as the DFT-level calculations confirm (Table 2), resulting in an apparent blue shift of the C=O stretching band.

Peaks at 1174 and 1388 cm^−1^ observed in solutions of PBMA were specific for PBMA because they were not observed for malic acid. The spectrum of PBMA did not change appreciably at pH 3, but at pH 4 the 1740 cm^−1^ peak was noticeably smaller and peaks at 1609 and 1401 cm^−1^ became prominent (Figure 3c). At pH 7 the remaining peak at 1736 cm^−1^ was due to ester C=O stretching and peaks at 1577 and 1393 cm^−1^ were observed, corresponding to carboxylate C=O stretching (Figure 3c, Table 1). Hence, the effect of pH on double hydrogen bonding in PBMA was similar to the effects on malic, acetic, and lactic acids. The PBMA peak at 1174 cm^−1^ remained when the pH increased, indicating that it was ester specific. On the basis of IR spectroscopy, the pKa for PBMA was ~3.5, in accord with previous measurements of 3.45–3.6 [24]. In conclusion, stronger double hydrogen bonds between carboxyl side chains were present in a moderately acidic solution of PBMA than in malic and lactic acids.

Quantum chemical calculations on the IR spectra of fragments approximating hydrogen-bonded and free PBMA (see also Figure 7, below), which due to the computational cost limitations were performed only at harmonic level, were again in agreement with the experiment. The doubly hydrogen-bonded group O-H stretching frequencies were predicted lower and C=O stretching ones higher than for lactic and malic acids (Table 2). The ester C=O stretching, however, was computed lower than that of the side-chain groups, contrary to the experiment. This again suggests the contribution of non-hydrogen bonded C=O vibrations to the band as an explanation for the higher than expected hydrogen-bonded side-chain vibrational frequency.

We also observed that the broad peak of absorbance by water centered at 2130 cm^−1^ due to a combination of bending and libration was shifted to lower frequencies in solutions of carboxylic acids at low pH (Figure 3, Table 1). This peak returned to about the same frequency as water itself when the pH was increased to produce carboxylates, showing that the shift was primarily due to carboxyl hydrogen bonding. The shift in frequency increased as the concentrations of malic and lactic acids and PBMA increased but the bending frequency was unchanged (Table 3), showing that the shift in the bending/libration frequency was due to a change in libration. This suggests that there is a significant contribution from interfacial water, whose libration modes are strongly influenced by the presence of the solutes. Since the bend/libration combination band shift with respect to bulk water occurs exclusively for doubly hydrogen-bonded carboxylic acids, the double hydrogen bonding must have an important effect on the interfacial water. The anharmonic vibrational calculations on lactic acid dimer with explicit water and a water cluster (Table 2), while yielding a broad bending/libration combination band in the 2300–1900 cm^−1^ region, do not reproduce the experimentally observed shift.

### 3.3. Double Hydrogen Bonding in Malic Acid, Lactic Acid, and PBMA is Robust

To determine whether the double hydrogen bonding in malic and lactic acids is disrupted at elevated temperatures, the effect of temperature on the FT-IR spectra was determined at 5 °C intervals from 0 to 85 °C over four hours. Absorbances due to malic and lactic acids decreased with increasing temperature. Subtraction of the absorbance due to water using the water peak at 1644 cm^−1^ as a reference allowed the maximum absorbance of the broad carboxyl O-H peak at 2600 cm^−1^ and the carboxyl C=O peak at 1724 cm^−1^ (malic acid) or 1728 cm^−1^ (lactic acid) to be measured relative to a minimum absorbance at 2170 cm^−1^. As the temperature increased, carboxyl O-H stretching band, measuring the amount of double hydrogen bonding, decreased as a percentage of the C=O stretching band intensity, measuring the total amount of carboxyl groups, by about one-third but was not eliminated (Figure 4). Consequently, double hydrogen bonding due to carboxyl groups remained in solution up to 85 °C. Chaotropic agents are often used to denature proteins or dissolve proteins isolated in inclusion bodies. Urea disrupts protein structure, but 6 M urea had only a slight effect on the spectrum of malic acid. Guanidine hydrochloride (6 M), which can form double ionic hydrogen bonds with carboxylates, did not eliminate the absorbance due to double hydrogen bonding (Table 1).

Similarly, we investigated whether melting of double-stranded regions of PBMA would increase with increasing temperature similar to melting of dsDNA segments. The temperature dependence of the FT-IR spectrum of PBMA in water at 464 μg/μL showed that absorbance due to carboxylic acid hydrogen bonding centered at 2550 cm^−1^ decreased up to 85 °C but was not eliminated, similar to malic and lactic acids (Figure 4). Hydrogen bonding absorbance at 2550 cm^−1^ as a proportion of total carboxyl absorbance decreased but with ratios at ~50% of the malic and lactic acid levels, as expected for a 1:2 ratio of COOH to C=O compared to 1:1 because half of the carboxyl groups of PBMA were involved in ester formation (Figure 4). A sharp transition corresponding to a melting temperature was not observed. Urea had little effect on the intensity of the carboxyl O-H stretching but reduced the frequency to 2535 cm^−1^. It also reduced the frequency of the water libration/bending mode (Table 1). Guanidine hydrochloride broadened and reduced the carboxyl O-H absorbance of PBMA (Table 1). These results showed that double hydrogen bonding in acidic solutions of malic and lactic acids and PBMA is robust.

### 3.4. PBMA Has Intramolecular Double Hydrogen Bonds

The ratio of absorbance due to carboxyl O-H stretching to absorbance due to C=O stretching did not change as the concentration of PBMA increased from 0.5 M to 4 M in carboxyl side chains, suggesting that hydrogen bonding involved primarily intramolecular interactions (Figure 5a).

If PBMA has double-stranded segments at low pH but not at neutral pH, size exclusion chromatography should detect a size difference even though there is a range of strand lengths. Intermolecular hydrogen bonding might lead to larger structures at low pH whereas intramolecular hydrogen bonding might lead to smaller structures due to folding. PBMA loaded at 10 mg/mL (90 mM in side chains) on a silica-based size exclusion column eluted over a range of volumes at pH 2.3 that was larger than at pH 7.2, showing that under acidic conditions PBMA was folded into intramolecularly hydrogen bonded structures that were smaller than the presumed open strands of neutral polyanionic PBMA (Figure 5b). By comparison with the elution volumes of protein standards, the elution volumes for maximum absorbances observed at pH 2.3 (9.4 mL) and 7.2 (7.8 mL) corresponded to those of proteins of ~25 kDa and ~130 kDa, respectively. Malic acid eluted at 11.0 mL at pH 2.3 and 10.5 mL at pH 7.2. Both the concentration dependence and size exclusion chromatography results showed that PBMA has intramolecular double hydrogen bonds.

### 3.5. A High Proportion of Fairly Rigid Double-Stranded Segments in PBMA

To obtain information about the structure of solid PBMA, it was examined by atomic force microscopy (AFM) [40]. After a section of mica was dipped in an aqueous solution of PBMA at 0.5 mg/mL and dried, AFM showed the presence of a layer or film of surface material with a height of ~0.8 nm (8 Å) (Figure 6a–c) in exact agreement with the diameter of ~7–8 Å predicted by density functional theory (DFT) calculations for double-stranded poly (β-L-malic acid) (Figure 7d). These layers seemed to be assemblies of PBMA molecules containing high proportions of double-stranded segments. PBMA was also observed as clumps. When PBMA solution was dropped on mica and allowed to dry, the clumpy form of PBMA was observed and layered material was not detected (Figure 6d). No layered or clumpy material was observed in the absence of PBMA. Layered PBMA seemed to form after rapid evaporation of a thin aqueous film.

Further structural information was obtained from circular dichroism (CD) spectra of PBMA in 100 mM potassium phosphate buffers at pH 2.2 (0.1 and 1.0 mg PBMA/mL, corresponding to 0.9 and 9 mM in side chains, respectively) and 7.2 (0.1 mg PBMA/mL, 0.9 mM in side chains) scanned from 185 to 260 nm. At both acidic and neutral pH the CD spectra of PBMA exhibit a strong couplet-like signal associated with the absorption of carboxylic side-chains (Figure 7a). The magnitude of the CD at pH 7.2 where hydrogen bonds are not present clearly demonstrated that even a single-stranded PBMA is not a featureless coil, but has locally well-defined conformation. Moreover, based on previously reported CD spectra of β-oligopeptides, the sign pattern suggests a right-handed helical conformation [41]. An identical but less intense CD observed at low pH where hydrogen bonding is present indicates that the CD is dominated by the non-hydrogen bonded single-stranded stretches, while the double-stranded segments have much weaker signals. This implies that double-stranded structures are more extended, quite rigid, and less chiral, not unlike, for example, peptide β-sheets.

**Figure 7 life-07-00035-f007:**
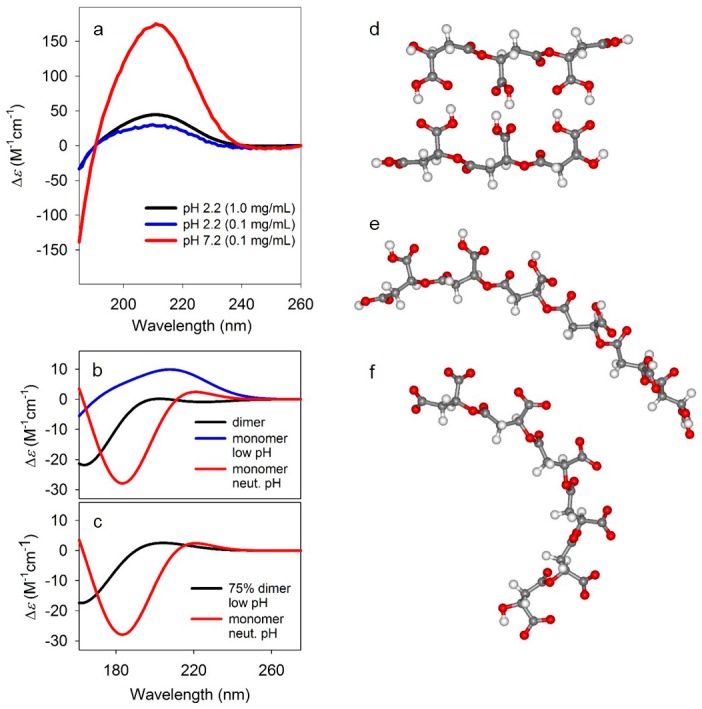
Circular dichroism spectra. (**a**) Experimental circular dichroism (CD) spectra PBMA at pH 2.2 (0.1 and 1.0 mg/mL) and 7.2 (0.1 mg/mL). (**b**) Simulated CD spectra for the dimer (black), monomer at low pH (blue), and monomer at neutral pH (red). The CD spectra simulations were carried out using time-dependent density functional theory (TD-DFT) CAM-B3LYP/6-31+G(d,p) level of theory with polarized continuum model (PCM) for the solvent (water). (**c**) Comparison of 75% dimer, 25% monomer spectrum at low pH, and the monomer spectrum at neutral pH. (**d**–**f**) Model structures for dimeric (**d**), and monomeric at low pH (**e**) and neutral pH (**f**) PBMA. The structures were optimized at B3LYP/6-31+G(d,p) level of theory in implicit water approximated by a polarized continuum model (PCM).

This hypothesis is confirmed by quantum mechanical simulations of CD spectra (Figure 7b,c) for model malic acid oligomers that mimic short segments of doubly hydrogen-bonded (Figure 7d) and non-bonded PBMA at both low (Figure 7e) and neutral (Figure 7f) pH. The oligomer structures were optimized at DFT B3LYP/6-31+G(d,p) level of theory [30,31,32] with implicit water [33] which was also used for the vibrational calculations described above (see Section 2, Table 2). The optimization yielded single non-hydrogen bonded strands (Figure 7e,f) with slight, but definitive right-handed helical twist, while the dimeric hydrogen-bonded structure did not have a discernable sense of left- or right-handed helicity. The CD spectra were computed for these optimized geometries at CAM-B3LYP/6-31+G(d,p) level [36] again with implicit solvent model [33] for water. The calculated CD for the monomeric oligomers in both protonated (acidic pH) and charged (neutral pH) forms qualitatively agreed with experiment, reproducing the observed sign pattern, confirming the locally right-handed helicity of the monomeric forms. By contrast, the computed CD for the dimeric, hydrogen-bonded oligomer (Figure 7b) are approximately opposite in sign and significantly less intense due to the more planar structure. For illustration, combining 75% of the dimeric with 25% of the protonated single strand (Figure 7c) yields approximately the same CD sign pattern to the charged oligomer and to experiment. While the agreement is not perfect, it is more than satisfactory given the highly simplified, short oligomer models, along with the known difficulties in accurately computing electronic CD spectra [42].

### 3.6. Ester Bonds Are Stable at Low Temperatures under Moderately Acidic Conditions

Ester bond hydrolysis has been extensively studied under strongly acidic, neutral, and basic conditions but not in dilute acids (pH > 1). Activated esters containing electron withdrawing groups may be more susceptible to hydrolysis by water under neutral conditions than acid catalyzed hydrolysis at low pH [43,44]. β-linked polyesters are more hydrolytically stable than α-polyesters and polyesters formed from a single stereoisomer are more stable than those from a racemic mixture [45]. PBMA is hydrolyzed in neutral and basic solution and in strongly acidic solution [24]. The closely related insoluble biological polyester, poly (β-L-hydroxybutanoic acid), which has methyl side chains, is hydrolyzed by alkali and concentrated H_2_SO_4_ but not by acid up to 4N [46]. These observations suggested that ester bonds might be stable under the acidic conditions where double hydrogen bonds between carboxyl groups are present.

We investigated the stability towards hydrolysis of the ester bonds of PBMA at pH 1.5 and 2.0, and two water-soluble esters, methoxyarginine (MeOArg) and methoxytyrosine (MeOTyr), at pH 2 over long periods. According to IR spectra, at 23 °C PBMA was largely unchanged after 48 h at pH 2 and pH 1.5 (Figure 8a). MeOArg has previously been reported to be unstable at neutral pH [47]. The peak at 1671 cm^−1^ for Arg side chain N-H stretching is a measure of Arg levels whether it is esterified or not. Relative to this N-H absorbance, the C=O stretching absorbance is substantially higher at pH 2 for 2 M MeOArg (83% at 1746 cm^−1^) than for 2 M Arg (27% at 1729 cm^−1^). Consequently, this ratio decreases during hydrolysis of MeOArg. After 35 days, little change was observed at 4 °C, a slight decrease was observed at 23 °C, and a larger decrease at 37 °C (Figure 8b). Hydrolysis of MeOTyr produces Tyr, which has low solubility. Tyr was first detected as a crystalline product from MeOTyr after 35 days at 4 °C, 28 days at 23 °C, and 14 days at 37 °C. These results demonstrated that ester hydrolysis was slow at low temperature under moderately acidic conditions.

## 4. Discussion

### 4.1. Double Hydrogen Bonding in PBMA Provides Support for the Poly (β-D-Malic Acid) Hypothesis

The discovery that PBMA dissolved in water has double hydrogen bonds between carboxyl groups provides critical experimental support for the hypothesis that poly (β-D-malic acid) was the original evolutionary predecessor of nucleic acids [3,21]. Malic acid is at the hub of biochemical metabolism and is therefore likely to have been synthesized early in the emergence of life. At this time a distinction may have been made between L-malic acid, which was used for metabolism, and D-malic acid, which was used to form poly (β-d-malic acid). It has not yet been established whether the hydrogen bonding interactions within PBMA are parallel, antiparallel, or a mixture of the two. However, because double hydrogen bonding is predominantly intramolecular, as implied by the lack of concentration dependence and size exclusion chromatography patterns (Figure 5), it is likely to be mostly antiparallel, similar to that found in RNA. Although the structure of PBMA is not known in detail, the CD spectrum of PBMA in water, together with the DFT calculations, indicate that the double-stranded segments of PBMA form a fairly rigid, extended structure. AFM of solid phase PBMA shows that it can form a film that has the height calculated for the diameter of a double-stranded polyester, suggesting that PBMA has a high proportion of its carboxyl groups involved in double hydrogen bonding. Consequently, PBMA seems to fold like RNA with considerable double-stranded content [48].

The change in O-H stretching frequency from ~2600 cm^−1^ in malic acid to ~2550 cm^−1^ in PBMA suggests that stronger hydrogen bonds form between carboxyl groups in the polyester than the monomer. It does not appear to be due to the cooperativity between individual hydrogen bonds [49], as we do not observe more cooperative melting transitions in PBMA than in the monomers (Figure 4). Rather, the reason may be lower solvent accessibility of the COOH groups in PBMA than in the monomers in analogy to proteins, where low polarity environment and less competition from solvent (water) molecules [50,51] are generally believed to lead to stronger hydrogen bonds [52,53,54]. We note, however, that some small molecule studies [55] find that hydrogen bonds are weaker in water than in solvents of lower polarity.

Chains with an average length of ~350 malic acid residues presumably contain segments involved in double hydrogen bonding and loops or turns that are single stranded. The fact that malic acid itself forms double hydrogen bonds shows that short segments could have been stabilized by these bonds, which would have been advantageous in primitive cells for formation of hydrogen bonds between polyester strands and between polyester strands and malic acid monomers used in polyester synthesis, such as malyl-thioesters.

### 4.2. Stability of Double-Stranded Hydrogen Bonding in PBMA

Double hydrogen bonding in PBMA is robust with regard to increasing temperature and the presence of chaotropic molecules. However, maximum strength double hydrogen bonding was observed in the low temperature range, 0–10 °C. The lack of a sharp transition in the temperature dependence experiment has two possible explanations. One is that the transition is cooperative, but occurs at higher temperature than could be experimentally accessed, and what we observe is just a gradual pre-melting. A more plausible explanation, supported by experimental and other studies of acetic acid [56,57,58,59], is that the enthalpy of dimerization is low, which would lead to a shallow melting curve as observed. Because nucleotides such as GMP and CMP do not form stable hydrogen-bonded base pairs in water [60], hydrogen bonding between carboxylic acids seems to be considerably stronger than pairing between purines and pyrimidines.

The broad absorbance band of water centered at 2130 cm^−1^ results from a combination of the bending frequency at 1644 cm^−1^ and a broad libration frequency calculated at 486 cm^−1^. Libration absorbance arises from motion of water molecules interacting with each other via hydrogen bonds. A substantial decrease in the frequency of the combination bending/libration absorbance is observed in solutions of carboxylic acids but not in solutions of carboxylates, suggesting that it arises from the double hydrogen bonding between carboxyl groups. The decrease in frequency is concentration dependent. Because the bending frequency is almost independent of carboxylic acid concentration, the change in bending/libration absorbance must be due to a decrease in the libration frequency. However, the concentration of dissolved carboxylic acids is low compared to the 55 M concentration of water and, therefore, the bending/libration absorbance of the majority of the water must be overlaid by a more intense but lower frequency absorbance due to a particular population of water molecules. Compared to water, libration in solutions of metal halides is generally increased in intensity and decreased in frequency, and is dependent upon the concentration and the nature of the cation and anion. Large anions produce a particularly large reduction in frequency [61]. Doubly hydrogen-bonded carboxylic acid dimers appear to behave like large halide anions with oxygen atoms in the dimers forming hydrogen bonds with O-H of water molecules in the primary hydration shell, as previously suggested for lactic acid [27], possibly creating a population of water molecules with decreased libration frequencies.

### 4.3. The Possibility that Carboxyl Group Pairing Predated Base Pairing

The essence of genetics is information propagation through complementary base pairing between strands of nucleic acids. Base pairing between strands is not entropically favorable. dsDNA stability is often discussed in terms of base pairing and base stacking. Melting temperatures for dsDNA are dependent upon base sequence, with triple hydrogen bonded GC base pairs increasing stability suggesting a role for hydrogen bonding in dsDNA stability [62,63]. On the other hand, stacking of bases at close to the optimal van der Waals distance is reported to make the dominant contribution to dsDNA thermal stability, with AT base pairing being destabilizing and GC base pairing having little effect [64]. Phosphate in the deoxyribose-phosphate backbones potentially destabilizes dsDNA through electrostatic repulsion, but solvation by water and cationic counterions for phosphate shield the negative charges from interacting with one another and contribute to DNA stability. Stable hydration of bases is observed in the grooves of dsDNA and is also involved in dsDNA stability [65,66]. Together these properties give dsDNA a stable and fairly rigid structure. RNA has a greater propensity to fold than ssDNA, with phosphate binding to metal ions and the 2′-OH of ribose playing prominent roles [48,67,68,69]. RNA is more hydrated than dsDNA due to the unpaired bases and the 2′-OH. As in DNA, stable hydration patterns are observed around double-stranded regions [70]. In our hypothesis for primitive cells, base stacking and phosphate would not be involved in the stability of double strands in the evolutionary predecessor of RNA and DNA because the biosynthetic pathways for purine and pyrimidine synthesis and for uptake and use of phosphate had not yet evolved. If this were the case, strong hydrogen bonding between side chains and supporting solvation effects would be required. Our results support the proposal that double hydrogen bonding between carboxylic acid side chains has these properties. Strong double hydrogen bonding in PBMA has been demonstrated and the significant shift in the bending/libration modes of water (Table 1 and Table 3, Figure 3) observed for doubly hydrogen-bonded carboxyl groups suggests a pronounced effect on the aqueous solvent. Presumably, this effect comes predominantly from the water molecules most directly involved in the interaction with these carboxyl groups (i.e., the closest solvation spheres).

### 4.4. Optimization of RNA and DNA as Genetic Polymers

Is RNA, a highly complex and hydrolytically unstable molecule, the best alternative for the origin of nucleic acids? Five of the reasons that have been provided in support of this idea are discussed below. First, RNA seems to be uniquely qualified to act as both a genetic polymer and a catalyst [71,72]. This may be correct, but there remains the unresolved problem of a feasible synthesis of RNA under presumed prebiotic conditions. An alternative to the RNA world hypothesis is that nucleic acids and proteins coevolved. Although it involves two polymers instead of one, the proposal that nucleic acid or nucleic acid predecessor synthesis was coupled to protein synthesis allows a simpler solution to the DNA/protein chicken and egg problem than synthesis of RNA alone. Ribosomal protein synthesis subsequently evolved from this coupled process [3,21,22].

Secondly, some coenzymes are thought to contain ‘vestiges’ of RNA as inactive components (such as adenosine, ADP) of a time when coenzymes were bound to RNAs [6,71,73]. An alternative explanation is that the adenosine/ADP components may have been useful for adapting nucleotide binding domains of proteins (e.g., Rossman fold) for other chemical reactions.

Thirdly, the peptidyl transferase center (PTC) of the ribosome where the peptide bond is formed is made from RNA. Therefore, RNA had to come before proteins. The idea that the PTC is a ribozyme was suggested by early crystal structure determinations of the ribosome [74]. However, RNA forming a PTC does not catalyze peptide bond formation [75], and the N-terminal tail of protein L27 is found in the PTC where it promotes peptide bond formation [76]. Thus, the PTC is now more difficult to describe as a ribozyme than it was previously, and may be better described as a ribonucleoprotein complex. Such a complex is consistent with coupled synthesis of proteins and nucleic acids preceding ribosomal protein synthesis [22].

Fourthly, the fact that PBMA does not contain phosphate in its backbone runs counter to the idea that a genetic polymer such as DNA requires a phosphate linkage (or more generally for a theoretical genetic polymer, a negative or positive charge in the backbone). One obvious reason for the presence of phosphate in nucleic acids is that it improves water solubility. Other reasons are that (a) electrostatic repulsion between phosphates forces double-stranded regions to hydrogen bond through Watson-Crick pairing rather than another type of hydrogen bonding such as Hoogsteen base pairing where the phosphate groups are closer, and b) phosphate discourages folding in DNA [77,78]. Although DNA can fold into complex structures, folding in RNA is more commonly found in biology where it is assisted by the presence of the 2′-OH group, and it is the 2′-OH group that makes RNA a more effective catalyst than DNA. The results with PBMA suggest that a negatively charged backbone may not be necessary when the hydrogen bonds involve carboxyl groups. While it is possible for the carboxyl side chains to form different types of hydrogen bonds with other carboxyl side chains and hydrogen bonds between carboxyl side chains and water probably occur in loops or turns between the double stranded regions of PBMA, cyclic double hydrogen-bonded structures between carboxyl groups are present in solution and in solid PBMA. Hydrogen bonding alternatives in nucleic acids, such as Watson-Crick and Hoogsteen base pairs, are not available for carboxyl groups. Therefore, a charged backbone was not necessary in the proposed original evolutionary predecessor of RNA.

Lastly, the presence of the ribose phosphate or deoxyribose phosphate backbones in nucleic acids plays an important role in genetics by ensuring that base pairing is complementary through base stacking at the right distance and in the right positions for van der Waals interactions with other bases. The backbone seems to be optimized for base stacking, supporting the RNA world position that it was required from the beginning. If, as we have surmised, ribose, phosphate, purines, and pyrimidines were not available initially, an alternative explanation is that the optimization of base stacking and the ribose phosphate backbone for genetic function accrued during the multistage evolution of the nucleic acids from a simple, but not optimal, genetic function involving carboxyl-containing side chains (as described in [3]).

### 4.5. Double Hydrogen Bonding between Carboxyl Side Chains on the Early Earth

An acidic environment is required for hydrogen bonding between carboxyl side chains. An acidity in the pH 2–3 range may not have been possible in the Hadean oceans, but volcanic land masses could have had pools that were acidified via fumaroles and rain out of high levels of CO_2_ and low levels of SO_2_, nitrogen oxides, and H_2_S released into the atmosphere through volcanic activity that was not too dissimilar from outgassing found in modern volcanoes [79,80,81,82,83,84]. Acidic pools having moderately acidic pH are found in geothermal regions on Earth [85,86]. Analysis of detrital zircons suggests the presence of granitic material on the early Earth that is indicative of continental crust [87,88], although an alternative interpretation has been proposed [89]. The predominant crustal rocks are thought to have been basaltic [90]. Whether or not volcanic activity allowed crustal material to form ‘dry land’ is a matter of dispute [10,91]. An acidic environment is more reasonably explained by the presence of dry land.

Evidence showing when life started on Earth is not firmly established in the geological record. It may have been in the Hadean eon some time following the formation of the Moon and before the late heavy bombardment [2,92]. The temperature of the Earth’s surface at that time is unknown [93], but zircons dated to this period indicate the presence of surface water [87,94]. Because the sun was 25–30% less luminous than it is now the oceans and surface water may have been mostly frozen (reviewed in [95]). A key unknown is the pressure of carbon dioxide in the atmosphere and the level of greenhouse gas activity that it provided during the Hadean [93].

### 4.6. Stability of Ester Bonds under Acidic Conditions

Formation of polyesters under acidic conditions seems to conflict with a well-known general method for hydrolyzing ester bonds that involves heating them with acids such as H_2_SO_4_ or HCl. Under moderately acidic conditions (pH 2), ester bonds are hydrolysed slowly, especially at low temperatures. PBMA undergoes rapid hydrolysis under strongly acidic conditions and is also hydrolysed in neutral solution where the polyanionic form is present [24]. According to FT-IR spectra, over a short period (48 hours) hydrolysis of PBMA at 23 °C is slow at both pH 2 and pH 1.5. Water-soluble esters such as MeOArg and MeOTyr are slowly hydrolyzed at pH 2 (Figure 8). Consequently, between strongly acidic and neutral conditions is a range of acidity, pH 2–3, where ester bonds are stable and double hydrogen bonds can form between carboxyl groups. PBMA contains polyester chains of variable lengths and single- and double-stranded segments may be expected in acidic solution where double hydrogen bonds are predominantly intramolecular. The dynamic nature of PBMA folding in water is unknown, and whether ester bonds in double-stranded segments are more stable to hydrolysis than those in single-stranded segments is also unknown. Presumably, carboxyl groups in single-stranded regions of PBMA are preferentially ionized over those engaged in double hydrogen bonds and are mainly responsible for the acidity of PBMA. Although we have shown that ester bonds are stable at pH 2 over the short term, it still needs to be determined whether they are stable enough to be used in a genomic molecule. The stability of the ester bond to hydrolysis is unlikely to approach that of DNA at approximately neutral pH. Indeed, increased hydrolytic stability may have been one of the main driving forces that led to the evolution of DNA. A low temperature for the early Earth’s surface water would befit ester bond stability in primitive cells.

### 4.7. From PBMA to a Simple Information Molecule

PBMA and hypothetical poly (β-D-malic acid) are not information polymers because at least two different side chains are required. Modification of some of the side chain carboxyl groups with another carboxyl-containing molecule such as β-alanine (Figure 1, Table 1) could have produced an information polymer with two different side chains because carboxyl groups from both modified and unmodified side chains could form hydrogen bonds in the same direction relative to the polyester backbone and ‘carboxyl pair’ with complementary strands. The robustness of the hydrogen bonding in monomeric carboxylic acids and PBMA suggests that such carboxyl pairing is possible, but it needs to be experimentally demonstrated. As the aqueous environment and the intracellular conditions of early cells became less acidic, carboxyl side chains would have been unable to preserve hydrogen bonding, as shown by the pH dependence of the O-H and C=O stretching absorbances. Replacement of ionizable carboxyl side groups by carboxamide has been proposed as the first step in the evolutionary pathway that changed side chains into purines and pyrimidines [3]. Under conditions closer to neutral pH, phosphodiester linking in the backbone could have allowed greater polymer stability towards hydrolysis [3,21].

## 5. Conclusions

The difficulty in understanding how nucleic acids became the genetic polymers can be surmounted if it is assumed that they evolved from a simple polymer. The results presented here for PBMA provide the first experimental support for the hypothesis that this polymer was poly (β-D-malic acid). We have shown through IR and DFT calculations that a simple linear polyester made from a molecule that is central to biological metabolism forms double hydrogen bonds in aqueous solution through its carboxyl side chains. Studies by AFM indicate that solid PBMA has a high proportion of double-stranded segments. Size exclusion chromatographic and concentration studies show that the double hydrogen bonds are intramolecular. Although the polymer has considerable conformational flexibility, double-stranded segments seem to form extended, fairly rigid, structures rather than flexible coils. Furthermore, we have demonstrated that these double hydrogen bonds are thermo-stable, and that ester bonds are stable for short periods at the low pH required for double hydrogen bonding. Finally, we have suggested model structures for the monomeric and dimeric forms consistent with the spectroscopic and microscopic experimental data. The next step is to show that modification of the carboxyl side chains with another carboxyl-containing molecule converts the polyester into a simple information polymer that forms double hydrogen bonds between modified and unmodified side chains.

## Figures and Tables

**Figure 1 life-07-00035-f001:**
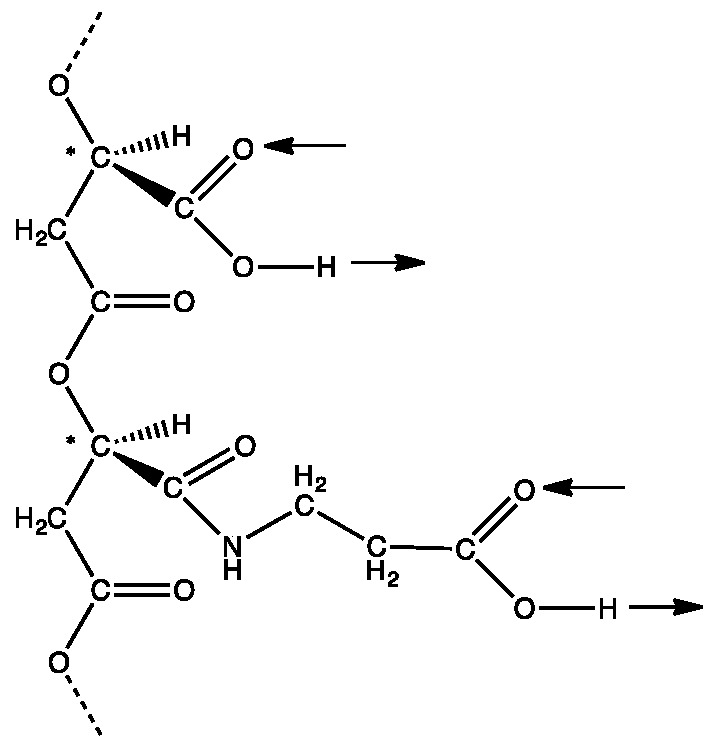
The hypothetical first information polymer with an ability to form double hydrogen bonds through side chain carboxyl groups. A section of poly (β-D-malic acid) containing two side chains, one with an unmodified carboxyl group and the other with the carboxyl group modified by formation of an amide link with β-alanine. These side chains can form complementary ‘carboxyl pairs’. Arrows show their potential for double hydrogen bonding in the same direction relative to the backbone. Chiral carbon atoms marked with *.

**Figure 2 life-07-00035-f002:**
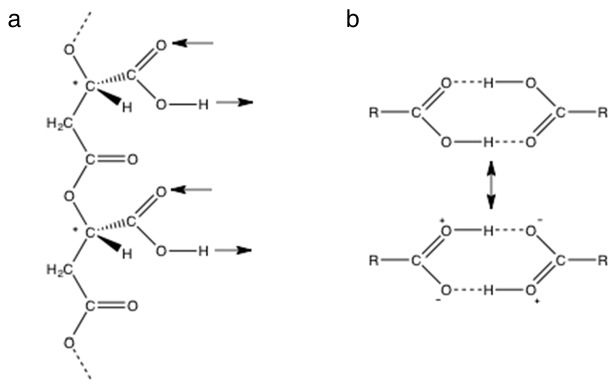
Hydrogen bonding by carboxyl groups. (**a**) A section of poly (β-L-malic acid) containing two side chain carboxyl groups with arrows showing their potential for double hydrogen bonding in the same direction relative to the backbone. Chiral carbon atoms marked with *. (**b**) Cyclic double hydrogen bonding between carboxyl groups of a carboxylic acid stabilized by an ionic resonance structure.

**Figure 3 life-07-00035-f003:**
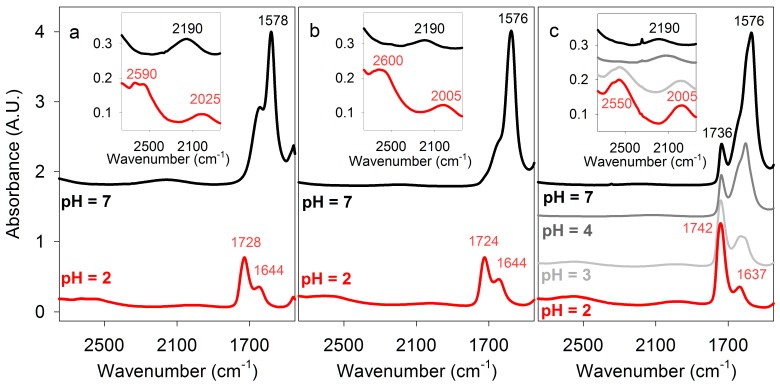
Fourier Transform infrared (FT-IR) spectra of carboxylic acids. Some absorbance maxima are indicated. (**a**) Lactic acid in H_2_O at pH 2 and 7. (**b**) Malic acid in H_2_O at pH 2 and 7. (**c**) Poly (β-D-malic acid) PBMA in H_2_O at pH 2, 3, 4, and 7. Inserts show enlargements of the 2700–1900 cm^−1^ spectral region, comprising double hydrogen-bonded carboxyl OH-stretching and H_2_O bending/libration modes.

**Figure 4 life-07-00035-f004:**
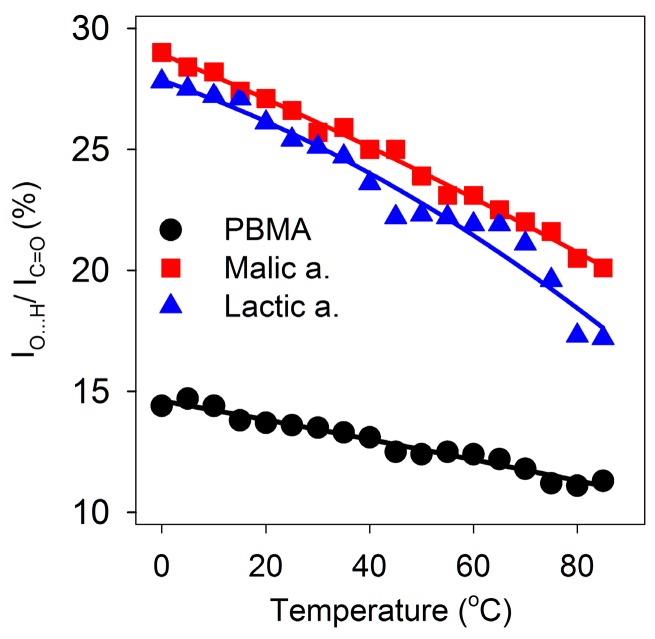
Dependence of double hydrogen bonding on temperature. O-H stretching absorbance at ~2600 cm^−1^ as a percentage of C=O stretching absorbance at ~1720 cm^−1^ from 0 to 85 °C for malic and lactic acids and PBMA.

**Figure 5 life-07-00035-f005:**
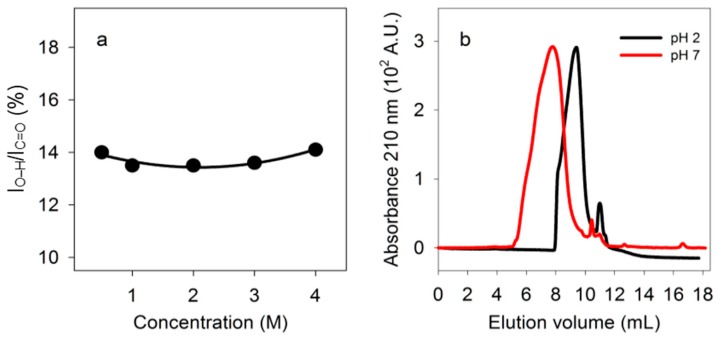
Tests for intramolecular double hydrogen bonds. (**a**) Dependence of double hydrogen bonding of PBMA on concentration. Percentage, calculated as in Figure 3, for different concentrations of PBMA carboxyl side chains in water at 23 °C. (**b**) Size exclusion chromatography. Fractionation of PBMA by size exclusion chromatography at pH 2.3 and 7.2.

**Figure 6 life-07-00035-f006:**
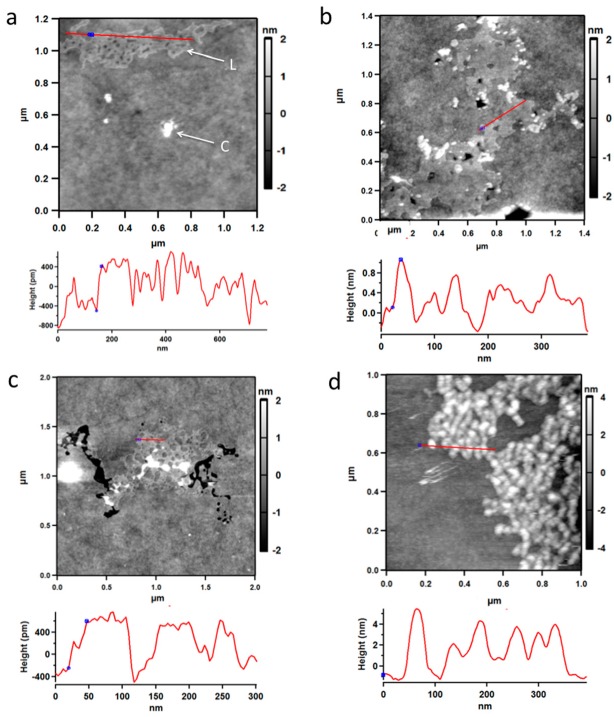
Atomic force microscopy of PBMA evaporated on mica from aqueous solution. (**a**–**c**) Mica dipped in aqueous PBMA and dried. Layered material with a height of ~8 Å (grey) is marked L and clumpy material (white) is marked C. (**d**) Clumpy material from PBMA solution dropped onto mica and dried. Cross sections corresponding to red lines on the figures show the heights of the material on the mica. Blue spots on the figures are also shown on the cross-sections. Black areas are blemishes in the mica.

**Figure 8 life-07-00035-f008:**
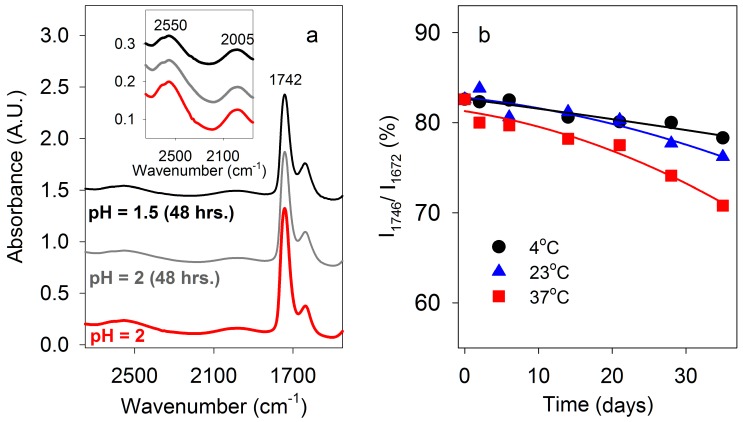
Stability of ester bonds under low pH conditions. (**a**) FT-IR spectra of PBMA at pH 2 after 0 and 48 h, and pH 1.5 after 48 h. (**b**) Time course of hydrolysis of 2 M ArgOMe at pH 2. FT-IR C=O absorbance at 1746 cm^−1^ as a percentage of the Arg side chain N-H absorbance at 1671 cm^−1^ determined at 4 °C, 23 °C, and 37 °C over 35 days.

**Table 1 life-07-00035-t001:** FT-IR absorbance frequencies (cm^−1^) of water, carboxyl, and carboxylate groups. Molarity of PBMA refers to concentration of carboxyl/carboxylate side chains. Values given for water bend/libration are before (+H_2_O) and after water subtraction (−H_2_O). NR = not resolvable.

	Carboxyl OH Str.	Carboxyl C=O Str.	Carboxylate COO^−^ Str.	Water Bend/Libration	
				+H_2_O	−H_2_O
H_2_O	-	-	-	2130	-
D_2_O	-	-	-	1560	-
Malic acid, 2 M, pH 1.6	2600	1724	-	2095	2005
Malic acid, 4 M, pH 1.25	2600	1724	-	2050	2000
Malic acid D_2_O	2600	1728	-	NR	NR
Malic acid 20% glycerol	2600	1724	-	2084	2035
Malic acid 20% glucose	2600	1724	-	2083	2015
Malic acid 20% galactose	2600	1724	-	2076	2000
Malic acid, urea 6 M	2600	1722	-	2063	2990
Malic acid, GuHCl 6 M	2600	1724	-	2067	2005
Malic acid, 2 M, pH 7	-	-	1576/1399	2144	2190
Lactic acid, 2 M, pH 2	2590	1728		2114	2025
Lactic acid, 2 M, pH 7	-		1581/1417	2152	2190
Acetic acid, 2 M, pH 2	2623	1712		2110	2030
Acetic acid, 2 M, pH 9	-	-	1555/1414	2135	2180
β-Alanine, 2 M, pH 3	2650	1720	-	2115	2010
β-Alanine, 2 M, pH 7	-	-	1574/1410	2128	2100
PBMA, 2 M, pH 2	2550	1742	-	2096	2005
PBMA, 4 M, pH 1.5	2555	1742	-	2045	2000
PBMA, 2 M, urea 6 M	2535	1739	-	2049	1965
PBMA, 2 M, GuHCl 6 M	~2560	1740	-	2049	1985
PBMA, 2 M, pH 7	-	-	-	2139	2190

**Table 2 life-07-00035-t002:** Computed vibrational frequencies for carboxyl groups in model compounds. The calculations were performed at B3LYP/6-31+G(d,p) level of theory with polarized continuum model (PCM) for water.

	Carboxyl O-H Stretch		Carboxyl C=O Stretch		H_2_O Bend/Libration
	harmonic	anharmonic	harmonic	anharmonic	anharmonic
lactic acid monomer	3734	3557	1773	1743	-
lactic acid dimer	3106 (−) *^a^* 3036 (+)	2686 (−) 2620 (+)	1712 (−) 1677 (+) *^b^*	1672 (−) 1623 (+) *^b^*	-
malic acid monomer	3733 (C_4_) *^c^* 3732 (C_1_) *^c^*	3520 (C_4_) *^c^* 3567 (C_1_) *^c^*	1778 (+) 1769 (−)	1743 (+) 1737 (−)	-
malic acid dimer	3119 (−) 3048 (+)	2680 (−) 2578 (+)	1722 (−) 1688 (+)	1684 (−) 1624 (+)	-
lactic acid dimer + 6 H_2_O	3107 (−) 3038 (+)	2689 (−) 2558 (+)	1690 (−) 1636 (+)	1656 (−) 1576 (+)	2300–1900
tri-malic acid dimer *^d^*	3098 (−) 3090 (−) 3083 (−) 3081 (−) 3075 (−) 3071 (−) *^e^* 3003 (+) 2987 (+) 2950 (+)	-	1739 (−) 1734 (−) 1726 (−) 1698 (+) 1694 (+) 1684 (+)	-	-
malic acid hexamer *^d^*	3734 3734 3733 3733 3733 3732	-	1803 1803 1802 1801 1800 1781 *^f^*	-	-
6 H_2_O	-	-	-	-	2320–1880

*^a^* In-phase modes are denoted by a (+), out-of-phase ones by a (−); *^b^* The (+) C=O stretching mode has negligible IR intensity; *^c^* Refer to C_1_ or C_4_ carboxyl groups of malic acid; *^d^* Ester C=O stretching modes 1782–1769 cm^−1^; *^e^* O-H stretching mixed with C-H stretching; *^f^* Contain contributions of ester C=O stretching modes; 1781–1769 cm^−1^.

**Table 3 life-07-00035-t003:** Dependence of the bending/libration and bending absorption frequencies of water (cm^−1^) on carboxylic acid concentration at low pH. Molarity of PBMA refers to the concentration of carboxyl side chains. Values for bending/libration obtained after water subtraction, except for water itself.

Molarity	0	0.5	1	2	3	4
*Bending/libration*						
Malic acid	2130	2050	2020	2005	2000	2000
Lactic acid	2130	2040	2030	2025	2025	2025
PBMA	2130	2080	2055	2005	1995	2000
*Bending*						
Malic acid	1644	1644	1644	1644	1644	1644
Lactic acid	1644	1644	1644	1645	1645	1645
PBMA	1644	1644	1643	1643	1642	1641

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
