# Peer review of "Double Hydrogen Bonding between Side Chain Carboxyl Groups in Aqueous Solutions of Poly (β-L-Malic Acid): Implication for the Evolutionary Origin of Nucleic Acids"

_life, 2017, doi:10.3390/life7030035_

Round 1

Reviewer 1 Report

This manuscript extends an idea first proposed in a long and thoughtful theoretical treatment by the first author, articulating the notion that D-malic acid polymers played a central role in the pre-biotic genesis of informational polymers. I reviewed that article enthusiastically, and find that this article evokes a similar enthusiasm. The primary source of my enthusiasm was the imaginative way in which the author developed his argument, that it was more or less independent of other proposals, and that it provided novel answers to quite basic questions.

The central idea is that poly(β-D-malic acid) (PBMA) served as an ancestral polymer that eventually became a rudimentary informational molecule via the incorporation of modifications of its carboxylate side chains that preserved the ability to form antiparallel pairs of double hydrogen bonds. There is a clever analogy here between such double hydrogen bonding and conventional base-pairing between purines and pyrimidines. Indeed, the authors suggest an evolutionary path between antiparallel chains of PBMA and nucleic acids in which the backbone carbon atoms of the poly malate were precursors to ribose-phosphate (with the eventual replacement of the C4 carbon by phosphate) and the side chain carboxylates were precursors to purines and pyrimidines. This proposal actually provides novel solutions for several puzzles about how informational polymers evolved.

The present manuscript describes biophysical—Fourier transform IR, circular dichroism spectroscopy and atomic force microscopy—and quantum-mechanical computational results supporting the formation at low pH of sets of “double hydrogen bonds” characteristic of antiparallel double helices in L isomeric form of PBMA. These results appear to provide important empirical evidence for the viability of the original hypothesis.

The evidence provided leave many unanswered questions. Nevertheless, the present paper is well constructed, clearly written, authoritatively documented, and thus can contribute materially to the resurgent dialog over how pre-biotic chemistry gave rise to biology. It should therefore merit publication in MDPI Life.

Concerns needing to be addressed:

(i) One puzzling aspect of the data and its interpretation that I feel the authors should address concerns the question of why their measurements show so little evidence of the cooperativity expected from the hydrogen bonds illustrated schematically in Fig. 2. Specifically, such behavior is notably absent from Fig. 3c, whereas I would have expected a quite dramatic, very much sharper transition over a much smaller pH range. Likewise in Fig. 4, I would have expected a more cooperative temperature dependent melting, and in Fig. 5a a more cooperative concentration dependence.

(ii) I disagree with the contention on page 8, lines 238-241 that the reduced solvent accessibility would result in “stronger hydrogen bonds”. I believe that this statement reflects a misunderstanding of the sources of strength in hydrogen bonding. If the authors disagree, then they should provide documentary evidence in references.

(iii) I may have missed this, but I do not understand why the simulated CD spectra in Fig. 7b and 7c have been plotted for different frequency ranges than the experimental signals in Fig. 7a. This discrepancy makes it difficult to understand the point the authors are making with respect to this figure.

Author Response

This manuscript extends an idea first proposed in a long and thoughtful theoretical treatment by the first author, articulating the notion that D-malic acid polymers played a central role in the pre-biotic genesis of informational polymers. I reviewed that article enthusiastically, and find that this article evokes a similar enthusiasm. The primary source of my enthusiasm was the imaginative way in which the author developed his argument, that it was more or less independent of other proposals, and that it provided novel answers to quite basic questions.

The central idea is that poly(β-D-malic acid) (PBMA) served as an ancestral polymer that eventually became a rudimentary informational molecule via the incorporation of modifications of its carboxylate side chains that preserved the ability to form antiparallel pairs of double hydrogen bonds. There is a clever analogy here between such double hydrogen bonding and conventional base-pairing between purines and pyrimidines. Indeed, the authors suggest an evolutionary path between antiparallel chains of PBMA and nucleic acids in which the backbone carbon atoms of the poly malate were precursors to ribose-phosphate (with the eventual replacement of the C4 carbon by phosphate) and the side chain carboxylates were precursors to purines and pyrimidines. This proposal actually provides novel solutions for several puzzles about how informational polymers evolved.

The present manuscript describes biophysical—Fourier transform IR, circular dichroism spectroscopy and atomic force microscopy—and quantum-mechanical computational results supporting the formation at low pH of sets of “double hydrogen bonds” characteristic of antiparallel double helices in L isomeric form of PBMA. These results appear to provide important empirical evidence for the viability of the original hypothesis.

The evidence provided leave many unanswered questions. Nevertheless, the present paper is well constructed, clearly written, authoritatively documented, and thus can contribute materially to the resurgent dialog over how pre-biotic chemistry gave rise to biology. It should therefore merit publication in MDPI Life.

Concerns needing to be addressed:

(i) One puzzling aspect of the data and its interpretation that I feel the authors should address concerns the question of why their measurements show so little evidence of the cooperativity expected from the hydrogen bonds illustrated schematically in Fig. 2. Specifically, such behavior is notably absent from Fig. 3c, whereas I would have expected a quite dramatic, very much sharper transition over a much smaller pH range. Likewise in Fig. 4, I would have expected a more cooperative temperature dependent melting, and in Fig. 5a a more cooperative concentration dependence.

We would like to thank the Reviewer for critical reading of the manuscript and for his insightful comments. 

Concerning the pH dependence the transition occurs between pH 3 and pH 4 exactly as expected for a reported pKa of ~3.5 and in accordance with the standard titration profile (Henderson-Hasselbalch equation). The lack of a sharp transition in the temperature dependence experiment has two possible explanations: first is that the transition occurs beyond the experimental temperature range and second, more likely, that the enthalpy of dimerization has a very low enthalpy as found by studies of e.g. acetic acid (Nash and Monk, 1957; Yamamoto and Nishi, 1990; Schrier et al., 1964; Chen et al., 2008) and which would lead to a shallow melting curve as observed. We have mentioned the lack of a sharp transition in the Results (p10):

            “A sharp transition corresponding to a melting temperature was not observed.”

 and the Discussion (p15):

            “The lack of a sharp transition in the temperature dependence experiment has two possible explanations. One is that the transition is more cooperative, but occurs at higher temperature than could be experimentally accessed and what we observe is just a gradual pre-melting. A more plausible explanation, supported by experimental and other studies of acetic acid (Nash and Monk, 1957; Yamamoto and Nishi, 1990; Schrier et al., 1964; Chen et al., 2008), is that the enthalpy of dimerization is low, which would lead to a shallow melting curve as observed.”

 The lack of concentration dependence seems to reflect the fact that the hydrogen-bonding is primarily intra-molecular, as also discussed in the manuscript.

 (ii) I disagree with the contention on page 8, lines 238-241 that the reduced solvent accessibility would result in “stronger hydrogen bonds”. I believe that this statement reflects a misunderstanding of the sources of strength in hydrogen bonding. If the authors disagree, then they should provide documentary evidence in references.

Studies of the effect of changing solvents on hydrogen bond strength in model molecules and proteins have, with occasional exceptions (Zheng et al., 2017), shown that hydrogen bond strength is weaker in water than in lower polarity environments (Klotz and Franzen, 1962; Ben-Tal, et al., 1997; Guo and Karplus, 1994; Cook et al., 2007; Gao et al., 2009; Bowie, 2011). We have discussed this in the revised manuscript (p15):

The change in O-H stretching frequency from ~2600 cm-1 in malic acid to ~2550 cm-1 in PBMA suggests that stronger hydrogen bonds form between carboxyl groups in the polyester than the monomer. It does not appear to be due to the cooperativity between individual hydrogen bonds (Guo and Karplus, 1994), as we do not observe more cooperative melting transitions in PBMA than in the monomers (Fig. 4). Rather, the reason may be lower solvent accessibility of the COOH groups in PBMA than in the monomers in analogy to proteins, where low polarity environment and less competition from solvent (water) molecules (Cook et al., 2007; Klotz and Franzen, 1962) are generally believed to lead to stronger hydrogen bonds (Bowie, 2011; Gao et al., 2009; Ben-Tal, et al., 1997). We note, however, that some small molecule studies (Zheng et al., 2017) found that hydrogen bond is weaker in water than in solvents of lower polarity.”

(iii) I may have missed this, but I do not understand why the simulated CD spectra in Fig. 7b and 7c have been plotted for different frequency ranges than the experimental signals in Fig. 7a. This discrepancy makes it difficult to understand the point the authors are making with respect to this figure.

Computed spectral frequencies (wavelengths) are not quantitatively accurate, due to well-known systematic (as well as and other) errors in approximate quantum chemical methods, such as the density functional theory used here. As a result, absolute wavelengths cannot be compared but the shape of the spectrum can. We could have shifted them to correspond to experiment, as is sometimes done, but we prefer not to do so as it introduces arbitrary correction factors and may lead to a false impression about the (absolute) accuracy of the DFT calculations.

References added:

Nash, G. R.; Monk, C. B. The Molecular Association of Some Carboxylic Acids in Aqueous Solution from e.m.f. Measurements.  J Chem Soc  1957, 4274-4280.

Yamamoto, K.; Nishi, N. Hydrophobic Hydration and Hydrophilic Interaction of Carboxylic Acids in Aqueous Solution: Mass Spectrometric Analysis of Liquid Fragments Isolated as Clusters. J. Am. Chem. Soc. 1990, 112,  549-558.

Schrier, E. E.; Pottle, M.; Scheraga, H. A. The Influence of Hydrogen and Hydrophobic Bonds on the Stability of the Carboxylic Acid Dimers in Aqueous Solution. J. Am. Chem. Soc. 1964, 86, 3444-3449.

Chen, J.; Brooks C. L. III; Scheraga, H. A. Revisiting the Carboxylic Acid Dimers in Aqueous Solution: Interplay of Hydrogen Bonding, Hydrophobic Interactions, and Entropy. J. Phys. Chem. B, 2008, 112, 242-249.

Zheng, V-Z.; Zhou, Y.; Liang, Q.; Chen, D-F.; Guo, R.; Xiong, C.-L.; Xu, Y.-J.; Zhang, Z.-N.; Huang, Z.-J. Solvent Effects on the Intramolecular Hydrogen-bond and Anti-oxidative Properties of Apigen: A DFT Approach. Dyes Pig. 2017, 141, 179-187.

Klotz, I. M.; Franzen, JS. Hydrogen Bonds between Model Pepide Groups in Solution. J. Am. Chem Soc. 1962, 84, 3461-3466. 

Ben-Tal, N.; Sitkoff, D.; Topol I. A.; Yang, A.-S.; Burt, S. K., Honig, B. Free Energy of Amide Hydrogen Bond Formation in Vacuum, in Water, and in Liquid Alkane Solution. J. Phys. Chem. B 1997, 101, 450-457.

Guo, H.; Karplus, M. Solvent Influence on the Stability of the Peptide Hydrogen Bond: A Supramolecular Cooperative Effect. J. Phys. Chem. 1994, 98, 7104-7105.

Cook, J. L.; Hunter, C. A.; Low, C. M. R.; Perez-Velasco, A.; Vinter, J. G. Solvent Effects on Hydrogen Bonding. Angew. Chem. Int. Ed. 2007, 46, 3706-3709.

Gao, J.; Bosco, D. A.; Powers, E. T.; Kelly, J. W. Localized Thermodynamic Coupling between Hydrogen Bonding and Microenvironment Polarity Substantially Stabilizes Proteins. Nat. Struct. Biol. 2009, 16, 684-691. 

Bowie, J. U. Membrane Protein Folding: How Important Are Hydrogen Bonds. Curr. Opin. Struct. Biol. 2011, 21, 42-49.

Reviewer 2 Report

The manuscript presents evidence that the carboxylated polyester PBMA (poly(b-D-malic acid) forms stable secondary structure in acidic aqueous solution through double hydrogen-bond interactions between free carboxylic acid side chains (primarily intramolecular). Supporting data includes IR and CD, with accompanying computational support. The manuscript proposes that the structural similarities between this polymer and the backbone of modern nucleic acids may suggest an evolutionary link between them. The hypothetical framework for this link has been previously outlined (Life 2015, 5, 467-505, parts 17/18)

The authors report that their data is the first evidence for ordered secondary structure in carboxylated polyesters dissolved in the aqueous phase. Although the conditions are somewhat extreme (pH 2, 2-4 M monomer equivalent), this data is still a welcome addition to the field.

Experimental chemistry papers should not be expected to provide a complete holistic pathway from simple building blocks to functional cells; however, the conditions of the proposed experiments (acidic, high concentration), in addition to the implausibility of an abiotic/early biotic formation of a homochiral polymer of malic acid, make the link between this work and the origins of life very tenuous. Correspondingly, portions of the manuscript purporting to justify this link are so optimistic that they are likely to antagonize, rather than inform, the field.

Examples include:

 “… there is a plausible evolutionary pathway leading to D-ribose based RNA and DNA. Specifically, the three carbons in the backbone of poly(b-D-malic acid) evolved into the 3’, 4’ and 5’ carbons of D-ribose. Replacement of the C4 carboxyl group of malic acid by a hydroxymethyl group allowed linking by phosphate.  The carboxyl side chains evolved into the pyrimidines and purines.”

 “this hypothesis also effectively reconciles the ‘genetics first’ and ‘metabolism first’ approaches.”

Other notes:

Figure 1 and Figure 2 show much the same thing. The inclusion of a b-alanine in Figure 1 is not justified (it is not referred to in the text until page 8.)

Author Response

The manuscript presents evidence that the carboxylated polyester PBMA (poly(b-D-malic acid) forms stable secondary structure in acidic aqueous solution through double hydrogen-bond interactions between free carboxylic acid side chains (primarily intramolecular). Supporting data includes IR and CD, with accompanying computational support. The manuscript proposes that the structural similarities between this polymer and the backbone of modern nucleic acids may suggest an evolutionary link between them. The hypothetical framework for this link has been previously outlined (Life 2015, 5, 467-505, parts 17/18)

The authors report that their data is the first evidence for ordered secondary structure in carboxylated polyesters dissolved in the aqueous phase. Although the conditions are somewhat extreme (pH 2, 2-4 M monomer equivalent), this data is still a welcome addition to the field.

We thank the reviewer for the critical review of our manuscript and for the positive remarks.

The experimental condition of pH 2-3 is required for formation of double hydrogen bonds between carboxyl groups. The conditions on the early Earth under which life began are unknown. A variety of conditions are possible including volcanic pools with this pH, similar to pools on Earth today.

A high concentration of PBMA was used to demonstrate the presence of the double hydrogen bonds by FT-IR. It was not chosen because it represented the concentration of polymer in primitive cells, which is unknown.

Experimental chemistry papers should not be expected to provide a complete holistic pathway from simple building blocks to functional cells; however, the conditions of the proposed experiments (acidic, high concentration), in addition to the implausibility of an abiotic/early biotic formation of a homochiral polymer of malic acid, make the link between this work and the origins of life very tenuous. Correspondingly, portions of the manuscript purporting to justify this link are so optimistic that they are likely to antagonize, rather than inform, the field.

Examples include:

 “… there is a plausible evolutionary pathway leading to D-ribose based RNA and DNA. Specifically, the three carbons in the backbone of poly(b-D-malic acid) evolved into the 3’, 4’ and 5’ carbons of D-ribose. Replacement of the C4 carboxyl group of malic acid by a hydroxymethyl group allowed linking by phosphate.  The carboxyl side chains evolved into the pyrimidines and purines.”

 “this hypothesis also effectively reconciles the ‘genetics first’ and ‘metabolism first’ approaches.”

A plausible pathway from poly(b-D-malic acid) to RNA has been proposed by one of us [see Francis 2015]. The reviewer does not provide an argument why formation of poly(b-D-malic acid) is implausible. It would be a simple polymer formed from a molecule at the hub of metabolism where it can be found in the form of an activated thioester. The monomer is therefore likely to have been one of the earliest molecules to be synthesized in primitive cells and its polymerization is chemically plausible, though not yet experimentally demonstrated, which we do not claim. We have added lines to the text on p3 to make clear that this is a hypothesis that requires considerable experimental testing:

Obviously, this hypothesis requires considerable experimental testing and this report is the first step in this direction.”

 Concerning whether we are presenting too optimistic a connection to the origin of life, we note that there are two hypotheses to choose from regarding the origin and evolution of nucleic acids: either RNA was somehow produced on the primitive Earth and used as the genetic material in primitive cells or RNA evolved from a simpler information polymer. If one accepts the first of these hypotheses, RNA synthesis depended upon extracellular synthesis of activated nucleotides until the ability of cells to catalyze RNA synthesis had evolved.  It is generally assumed that primitive cells would not have had the ability catalyze all of the steps for RNA synthesis. Proponents of this hypothesis particularly point to the synthesis of nucleotides by the reactions reported by the Sutherland group (e.g. Patel et al., 2015, reference in the paper). While this is a remarkable organic synthesis, it relies upon a specific sequence of chemical reactions occurring in a mixture of chemicals - what is now being referred to as ‘messy chemistry’. Nucleotides are then used to make RNA using mineral surfaces and/or wet-dry cycles. We, and others, find this to be an overly optimistic scenario and are therefore left with the second hypothesis. If RNA has an evolutionary history, the obvious question is: what was the initial information polymer from which RNA evolved? A number of interesting polymers have been synthesized in which the backbone of RNA has been simplified while retaining the purine and pyrimidine side chains for stacking and base pairing. These polymers are somewhat simpler than RNA but they are far from simple because they still require synthesis of the bases. If the original polymer required only a few synthetic steps, its synthesis could have been intracellular. This makes evolution of the nucleic acids more easily understandable because the long metabolic pathway for RNA biosynthesis could evolve from synthesis of a simple polymer. From our point of view this is a plausible explanation for the origin of the nucleic acids, and more plausible than the RNA first hypothesis. We accept that there are steps in the evolutionary pathway from the simple polymer to RNA that need to be explored experimentally. 

Other notes:

Figure 1 and Figure 2 show much the same thing. The inclusion of a b-alanine in Figure 1 is not justified (it is not referred to in the text until page 8.)

Figure 1 was included at the beginning of the manuscript to show how poly(b-D-malic acid) could be the basis for an information polymer through modification of the carboxyl side chains. The reader can then understand how the study of poly(b-L-malic acid) is related to the origin of an information polymer.     

Reviewer 3 Report

     I have read the manuscript “Double Hydrogen Bonding between Side Chain Carboxyl Groups in Aqueous Solutions of Poly(β-L-malic acid): Implication for the Evolutionary Origin of Nucleic Acids” by Francis, Watkins, and Kubelka carefully and critically.  The fundamental hypothesis of this manuscript is that “the earliest evolutionary predecessor of nucleic acids was the simple linear polyester, poly(β-D-malic acid), for which the carboxyl side chains could form double hydrogen bonds.”  The authors then present work dealing with poly(β-L-malic acid- a closely related polyester.  Employing a combination of FT-IR spectroscopy, liquid chromatography employing a size exclusion HPLC column, CD spectroscopy, and DFT calculations, the authors put forth their position concerning the plausible evolutionary predecessor of nucleic acids.

The authors’ review of the literature places the work in the current manuscript in good perspective.

The reported experiments appear to be well done and their interpretation appears, for the most part, to be consistent with the experimental observations.

This reviewer believes that it is a big leap from the authors’ “polyester hypothesis” to nucleic acids.

The most serious issue is the stability of the ester functionality in an acidic aqueous environment.  Using IR spectroscopy, the authors conclude that the ester functionality in the polyester is stable.  These experiments were conducted over a period of only 48 hours.  It is difficult for this reviewer to conclude that this polyester is stable in an evolutionary context.  There should be at least some equivocation. 

Overall, the manuscript is well-written and thought-provoking.  While I believe the authors tend to overstate the pertinence of their experimental results with respect to their hypothesis, I believe the manuscript is worthy of publication. 

Author Response

I have read the manuscript “Double Hydrogen Bonding between Side Chain Carboxyl Groups in Aqueous Solutions of Poly(β-L-malic acid): Implication for the Evolutionary Origin of Nucleic Acids” by Francis, Watkins, and Kubelka carefully and critically.  The fundamental hypothesis of this manuscript is that “the earliest evolutionary predecessor of nucleic acids was the simple linear polyester, poly(β-D-malic acid), for which the carboxyl side chains could form double hydrogen bonds.”  The authors then present work dealing with poly(β-L-malic acid- a closely related polyester.  Employing a combination of FT-IR spectroscopy, liquid chromatography employing a size exclusion HPLC column, CD spectroscopy, and DFT calculations, the authors put forth their position concerning the plausible evolutionary predecessor of nucleic acids.

The authors’ review of the literature places the work in the current manuscript in good perspective.

The reported experiments appear to be well done and their interpretation appears, for the most part, to be consistent with the experimental observations.

This reviewer believes that it is a big leap from the authors’ “polyester hypothesis” to nucleic acids.

We thank the reviewer and appreciate the positive comments.

Indeed, many steps would be involved in the evolution of nucleic acids from poly(b-malic acid) but there are important reasons for beginning with a simple polyester:         

a) the monomers could be synthesized and activated intracellularly in primitive cells  allowing them to be independent of a supply of monomers from the environment,                                    b) the long biosynthetic pathway for RNA could evolve intracellularly from the simple polymer with improvement of genetic function,                                                                

c) the complicated process for uptake and utilization of phosphate need not have been present at the beginning.                                                                                                              

We intend to show that there is a pathway from polyesters to nucleic acids. This study is the first step in this endeavor.

The most serious issue is the stability of the ester functionality in an acidic aqueous environment.  Using IR spectroscopy, the authors conclude that the ester functionality in the polyester is stable.  These experiments were conducted over a period of only 48 hours.  It is difficult for this reviewer to conclude that this polyester is stable in an evolutionary context.  There should be at least some equivocation. 

It is surprising that the ester bond is fairly stable at pH 2 because the general impression is that esters are hydrolyzed under acidic conditions. The insoluble naturally occurring polyester, poly(b-L-hydroxybutanoic acid) is not hydrolyzed even by 4N H+ (Yu et al., reference in the manuscript).  We have only shown that ester bonds are stable over short periods and have not yet studied the long-term stability of PBMA at pH 2-3. We do not expect the stability of the ester bond to hydrolysis to approach that of DNA at approximately neutral pH.  Indeed, increased hydrolytic stability may have been one of the main driving forces that led to evolution of DNA. We have added words of equivocation on this issue on page 18:

Although we have shown that ester bonds are stable at pH 2 over the short term, it still needs to be determined whether they are stable enough to be used in a genomic molecule.  The stability of the ester bond to hydrolysis is unlikely to approach that of DNA at approximately neutral pH.  Indeed, increased hydrolytic stability may have been one of the main driving forces that led to evolution of DNA.”

Overall, the manuscript is well-written and thought-provoking.  While I believe the authors tend to overstate the pertinence of their experimental results with respect to their hypothesis, I believe the manuscript is worthy of publication.